# Encapsulation of Formosa Papaya (*Carica papaya* L.) Seed Extract: Physicochemical Characteristics of Particles, and Study of Stability and Release of Encapsulated Phenolic Compounds

**Mércia da Silva Mesquita** [1]**, Priscila Dayane de Freitas Santos** [1]**, Augusto Tasch Holkem** [2]**, Marcelo Thomazini** [1]
**and Carmen Silvia Favaro-Trindade** [1,*]

1    Departamento de Engenharia de Alimentos (ZEA), Faculdade de Zootecnia e Engenharia de
     Alimentos (FZEA), Universidade de São Paulo (USP), Pirassununga 13635-900, SP, Brazil
2    Department of Biomedical Sciences, Faculté de Médecine Vétérinaire, Université de Montréal,
     Saint-Hyacinthe, QC J2S 2M2, Canada
*    Correspondence: carmenft@usp.br

**Abstract:** Papaya seeds are a promising source of phenolic compounds, but these are unstable, and the papaya extract has a bitter taste. This study aimed to encapsulate papaya seed extracts at different maturation levels, and to characterize the obtained microparticles for their physicochemical properties, chemical stability and release of bioactives. Extracts of papaya (unripe and ripe) seeds were spray-dried using concentrations of 0, 15 and 30% of maltodextrin and inlet air temperatures of 130 and 150 °C. The powders were analyzed for yield, moisture, Aw, hygroscopicity, dispersibility, color parameters, morphology, mean diameter, total phenolics, antioxidant activity, stability during storage and release of phenolics in simulated gastrointestinal fluids. Powders produced with maltodextrin showed better results regarding particle diameter, hygroscopicity, dispersibility, and phenolic stability during storage. All powders showed antioxidant action and significant values of total phenolic compounds. Samples without maltodextrin underwent caking. Drying temperatures had little influence on the characteristics of the powders. Encapsulated phenolic compounds were released in large amounts in the intestinal phase (86.6–100%). Powders produced with unripe seeds, 15% of maltodextrin and an inlet air temperature of 130 °C showed the best results. Thus, encapsulation was efficient, and encapsulated papaya seed extract has potential for food application as a natural additive.

**Keywords:** antioxidants; phenolics; stability; digestibility; microcapsules; maltodextrin





## 1. Introduction

Papaya (*Carica papaya* L.) is a tropical fruit which is native to America but has been cultivated in different areas of the planet [1]. In 2019, the worldwide production of papaya reached 13 million tons, and Brazil is the third largest producer (1 million tons in 2019); India leads the world (5.7 million tons in the same year) [1]. Due to the process of turning papaya into candied fruit, raisins, nectars, jellies, juices, jams, papain and pectin extract, a large amount of agro-industrial waste is generated that may cause environmental organic pollution. In processing, peels and seeds are removed and discarded, which constitute about 50% of the total weight of the fruit, and the seeds correspond to an average of 14% [2,3].

A way to reduce the disposal of these agro-industrial residues in the environment is to use the seeds. Papaya seeds are a promising source of natural antioxidants, phenolics and bioactive compounds that could be useful for the food industry. Piovesan and Viera [4], when evaluating the effect of adding papaya seed extract to chicken sausage, observed that the addition of 1.5% of the extract to sausages was effective at reducing lipid oxidation compared to the control (no extract added). Papaya seed extracts can also be used to reduce the oxidation of vegetable oils and foods [5,6].

It is known that most bioactive compounds degrade over time, with exposure to heat, extreme pH, light, oxygen, or food processing, losing their activity. It is believed that this problem can be minimized with the use of microencapsulation technologies, which aim to avoid the interactions between the bioactive and environmental factors; promote the protection and conservation of sensitive food compounds during the processing and storage period; contribute to a longer storage period and better action in foods; facilitate controlled release in the bodies of those who ingest them or foods with them; mask or preserve flavors and aromas; and transform liquids into solid ingredients for easy handling, incorporation into formulations, storage and transport [7,8].

Several techniques are employed to form microcapsules or microspheres, including spray drying, a low-cost, well-known process which is used to produce dry powders, granules, or agglomerates by means of atomization in chambers of a liquid product in hot air stream to instantly obtain a powder [9,10]. This technique has the advantages of being easy to scale up, continuous and highly efficient; and, in previous works by our group, it has already been shown to be efficient for the encapsulation of plant extracts rich in phenolic substances, such as those obtained from jabuticaba peels [11], Bordo grape winemaking pomace [12], peanut peels [13] and cinnamon Ceylon [14].

Given the above, the use of papaya seeds in the production of extracts is a relevant alternative, because in addition to providing new ingredients for industries, it aims to reduce the volume of waste of this residue. Due to the lack of research on the microencapsulation of bioactive compounds from papaya seed extracts, this work is innovative, as we carried out a study on the encapsulation of unripe and ripe papaya seed extracts by spray drying using maltodextrin as a carrier agent; in addition, the particles were characterized, and their in vitro digestibility was also studied.

## 2. Materials and Methods

### 2.1. Materials

Unripe papaya seeds (maturation stage 0) of the Formosa variety (*Carica papaya* L.) were donated by the factories "Doces Caseiros Mineiro", located in Uberaba, Minas Gerais, Brazil; and "Doces Colmeia Ltd.a", located in Caldas, Minas Gerais, Brazil. Ripe papaya seeds (maturation stage 5) were donated by Doceria Schmidt Ltd.a, in the District of Engenheiro Schmitt in São José do Rio Preto, São Paulo, Brazil. Maltodextrin DE10 (MOR-REX®. 1910), kindly donated by Ingredion (Mogi-Guaçu, Brazil), was used as the carrier agent. Hexane, hydrochloric acid (37%) and ferrous sulfate were provided by Êxodo Científica (Sumaré, Brazil). Ethanol (99.5%), sodium carbonate ($Na_2CO_3$) and Folin–Ciocalteu reagent were purchased from Dinâmica (Indaiatuba, Brazil). Potassium persulfate, sodium acetate trihydrate and ferric chloride hexahydrate were provided by Synth (Diadema, Brazil). For simulated digestion assay, potassium chloride, magnesium chloride ($MgCl_2\cdot6H_2O$), sodium hydroxide, calcium chloride ($CaCl_2\cdot2H_2O$), potassium phosphate monobasic, sodium bicarbonate, sodium chloride and ammonium carbonate were acquired from Synth (Diadema, Brazil). Pepsin (from porcine gastric mucosa, ≥250 units/mg solid, EC 3.4.23.1), bile salts and pancreatin (from porcine pancreas, 8 x USP specifications, EC 232-468-9) were purchased from Sigma-Aldrich (St. Louis, MO, USA). ABTS reagent (>98%), Trolox standard (6-hydroxy-2,5,7,8-tetramethylchroman-2-carboxylic acid, 97%, code 218940050) and TPTZ ((2,4,6-tris(2-pyridyl)-s-triazine), ≥98%, code T1253) were also acquired from Sigma (Sigma-Aldrich, St. Louis, MO, USA). All reagents were analytical grade.

### 2.2. Production of Liquid Extracts

The unripe and ripe papaya seeds were washed to remove pulp residues. After washing, they were dehydrated in an oven with forced air circulation (Pardal, model PE 60, Pardal Tec, Petrópolis, Brazil) at 65 °C for 18 h. After drying, the dehydrated seeds were ground in a domestic blender and sieved in 20 mesh sieves.

Aiming at the production of a pure and rich-phenolics extract, the seeds were defatted. For the extraction of papaya seed oil, we used 1 part of dried and crushed papaya seeds to

5 parts of the solvent hexane; that is, 1:5 (*w/v*) This mixture was kept under mechanical agitation (model 713DS, Fisatom, São Paulo, Brazil) at 600 rpm for 30 min at room temperature. Subsequently, the solution was filtered through Whatman n° 3 filter paper with the aid of a vacuum pump (model 132-2VC, Prismatec, Itu, Brazil) to separate the seeds.

The filtered seeds were dried in an oven at 65 °C for 18 h to evaporate the hexane. Subsequently, the proportion 1:10 (*w/v*) was used, adding 1 part of dried and crushed papaya seeds (defatted) to 10 parts of solvent (40% *v/v* aqueous ethanol). The mixture was kept under mechanical agitation at 600 rpm in a water bath (MA127, Marconi, Piracicaba, Brazil) at 60 °C for 45 min. After this period, the extracts were centrifuged (5430 R, Eppendorf, Hamburg, Germany) at 25 °C and 7000 rpm for 5 min to separate the solids. The extract was filtered through Whatman No. 3 filter paper using a vacuum pump, and the filtered solution was concentrated in a rotary evaporator (TE-211, Tecnal, Piracicaba, Brazil) at 45 °C, reducing it to 40% of the volume produced after extraction.

### 2.3. Production of Powdered Extracts by Spray Drying

The pure extract and mixtures (extract and carrier in different proportions) were homogenized with an UltraTurrax (T25 digital, IKA Brasil, Campinas, Brazil) at 10,000 rpm for 3 min, and aliquots were removed to determine the total phenolic content and moisture. Spray drying was carried out in a pilot spray dryer model MSD 5.0 from Labmaq do Brasil Ltda. (Ribeirão Preto, Brazil), which operated with a 2.0 mm diameter injector nozzle, an airflow of 40 L/min and a pressure of 2 bar atomization. The feed rate of the mixture was 10 mL/min. A $2 \times 3$ factorial experiment was carried out according to a completely randomized design (DIC) for each type of seed (unripe and mature). The variables tested were inlet drying air temperatures of 130 and 150 °C and various carrier agent concentrations (0, 15, and 30% of maltodextrin in relation of extract—*w/w*), adding up to 6 treatments (Table 1) with two replications, making 12 tests in total, for each type of seed.

**Table 1.** Formulations and drying conditions of the samples.

| Treatments | Fruit Ripening Stage | Inlet Air Temperature (°C) | Carrier Agent (%) | Water (g) | Maltodextrin (g) | Extract (g) | Final Weight of the Solution (g) |
|---|---|---|---|---|---|---|---|
| V1 | Unripe | 130 | 0 | 0 | 0 | 300 | 300 |
| V2 | Unripe | 150 | 0 | 0 | 0 | 300 | 300 |
| V3 | Unripe | 130 | 15 | 34.6 | 34.6 | 230.8 | 300 |
| V4 | Unripe | 150 | 15 | 34.6 | 34.6 | 230.8 | 300 |
| V5 | Unripe | 130 | 30 | 0 | 69.2 | 230.8 | 300 |
| V6 | Unripe | 150 | 30 | 0 | 69.2 | 230.8 | 300 |
| M1 | Ripe | 130 | 0 | 0 | 0 | 300 | 300 |
| M2 | Ripe | 150 | 0 | 0 | 0 | 300 | 300 |
| M3 | Ripe | 130 | 15 | 34.6 | 34.6 | 230.8 | 300 |
| M4 | Ripe | 150 | 15 | 34.6 | 34.6 | 230.8 | 300 |
| M5 | Ripe | 130 | 30 | 0 | 69.2 | 230.8 | 300 |
| M6 | Ripe | 150 | 30 | 0 | 69.2 | 230.8 | 300 |

### 2.4. Characterization of Powders

The drying process yield was measured by dividing the dry weight of powder obtained by the dry weight of the mixture added before drying, according to Equation (1).

$$\text{Yield \%} = \frac{dw\ powder}{dw\ mixture} \times 100 \tag{1}$$

The moisture content was evaluated using infrared radiation using a moisture analyzer model MB-35 (Ohaus, Parsuppany, NJ, USA), and the result is expressed as a percentage. The water activity in the samples was determined using the AQUALAB equipment (Decagon Devices, Pullman, WA, USA).

### 2.4.1. Hygroscopicity

Hygroscopicity was determined according to the methodology described by Cai and Corke [15], with some adaptations. Petri dishes containing 0.2 g aliquots of the samples were placed in a desiccator containing saturated NaCl solution (75.2% RH) for one week in a chamber at 25 °C. Hygroscopicity was measured by the mass of water absorbed by the sample and expressed in grams of water absorbed per 100 g of dry matter.

### 2.4.2. Dispersibility

Dispersibility determination was performed using the gravimetric method, described by Eastman and Moore [16], with modifications by Cano-Chauca, Stringheta [17]. Then, 0.2 g of the samples were added in erlenmeyers containing 20 mL of distilled water, and they were homogenized using an orbital shaker table (model TE-420, TECNAL, Piracicaba, Brazil) at 100 rpm for 30 min at 25 °C. Subsequently, the solution was centrifuged (5430 R, Eppendorf, Hamburg, Germany) at 3000 rpm for 5 min at 25 °C. A 10 mL aliquot of the supernatant was removed and placed in an oven at 105 °C until constant weight. Dispersibility was determined based on the initial mass of the sample dispersed in the 10 mL aliquot of the supernatant. The result is expressed as a percentage.

### 2.4.3. Instrumental Color

The color of each sample was measured with a colorimeter (Mini Scan XE Model, Hunter Lab, Reston, VA, USA) and expressed using the CIELAB color system (L*, a*, b*). The equipment was calibrated using a white and a black color plate.

### 2.4.4. Average Particle Diameter

The average diameter of the particles was determined in a laser diffraction device (model SALD/201V, Shimadzu, Kyoto, Japan) with a measuring range between 0.5 and 500 μm. The dry powders were dispersed in ethanol, and each sample was subjected to three readings.

### 2.4.5. Particle Morphology

The morphology of the particles was obtained by scanning electron microscopy (SEM), using the TM 3000 model microscope, Tabletop Microscope Hitachi (Tokyo, Japan), with the TM 3000 program.

### 2.4.6. Determination of Total Phenolic Content

The total contents of phenolic compounds in the powders were determined by the Folin–Ciocalteu method described by Singleton and Orthofer [18], with adaptations. The powders were diluted in different concentrations. An aliquot of 0.25 mL of each diluted powder was added to 2 mL of distilled water and 0.25 mL of Folin–Ciocalteu reagent. The tubes were placed for 3 min in a light-free environment. Then, 0.25 mL of saturated sodium carbonate solution ($Na_2CO_3$) was added to the mixture, which was homogenized in a tube shaker (model AV-2, Gehaka, São Paulo, Brazil) for 10 s. The tubes were placed in a water bath in the dark at 37 °C for 30 min to complete the reaction. Absorbance was measured at 740 nm in a spectrophotometer (Genesys 10S UV-Visible, Thermo Fisher Scientific, Waltham, MA, USA). The result is expressed in mg of gallic acid/g of powder, calculated using a calibration curve using gallic acid as a standard.

### 2.4.7. Antioxidant Capacity by the ABTS•+ Method

The antioxidant activity of the powders was determined through the capture of the free radical ABTS•+, according to the methodology described by Rufino and Alves [19]. The ABTS•+ radical solution was produced by reacting 5 mL of the 7 mM ABTS•+ stock solution with 88 μL of the 140 mM potassium persulfate solution. The solution remained free from light and at room temperature for 16 h to stabilize the solution and complete the reaction. Subsequently, this solution was diluted in ethanol until obtaining an absorbance of 0.7 ± 0.05 at 734 nm. The powders were diluted in different concentrations. In a dark

environment, 30 μL of each powder dilution was transferred to a test tube, and 3 mL of ABTS•+ radical was added, which were homogenized in a tube shaker for 10 s. The tubes were kept in the dark for 6 min to complete the reaction, then the absorbance was measured at 734 nm in a spectrophotometer (Genesys 10S UV-Visible, Thermo Fisher Scientific, Waltham, MA, USA) using ethanol as a blank. The result was expressed in μM of Trolox⁄g of powder, calculated using a calibration curve using the Trolox reagent as the standard.

### 2.4.8. Antioxidant Capacity by Ferric Reducing Antioxidant Power (FRAP) Assay

The antioxidant activity of the powders was determined by Ferric Reducing Antioxidant Power (FRAP) assay using the methodology described by Rufino and Alves [20]. The FRAP reagent solution was produced by combining 25 mL of 0.3 M acetate buffer, 2.5 mL of TPTZ solution at 10 mM and 2.5 mL of 20 mM aqueous ferric chloride hexahydrate. The powders were diluted in different concentrations. In the dark, an aliquot of 90 μL of each powder dilution was transferred to a test tube, and 270 μL of distilled water and 2.7 mL of the FRAP reagent were added. The tubes were homogenized in a tube shaker for 10 s and kept in a dark environment and in a water bath at 37 °C for 30 min to complete the reaction. Subsequently, the absorbance was measured at 595 nm in a spectrophotometer (Genesys 10S UV-Visible, Thermo Fisher Scientific, Waltham, MA, USA) using the FRAP reagent as a blank. The result is expressed as μM of ferrous sulfate/g of powder, calculated using a calibration curve using the 2 mM ferrous sulfate reagent as a standard.

### 2.5. Stability of Phenolics during the Powders' Storage

Powder samples were stored as described by Tonon and Brabet [21]. In short, 0.6 g of each sample (powder) was stored in a glass vial and placed in a desiccator containing saturated magnesium chloride $MgCl_2$ solution (32.8% RH). The desiccators were kept at room temperature for 90 days. The samples were evaluated every 15 days in relation to the total phenolic content, as described in item 2.4.6.

The color analysis was performed at the end of the stability test, as described in item 2.4.3, and the total color difference (ΔE*) obtained through Equation (2) was evaluated.

$$\Delta E^* = [(\Delta L^*)2 + (\Delta a^*)2 + (\Delta b^*)2]1/2 \tag{2}$$

where: ΔL* = L* start time–L* end time; Δa* = a* start time–a* end time; Δb* = b* start time–b* end time.

### 2.6. Kinetics of Release of the Phenolic Compounds during "In Vitro" Digestion

In vitro digestion was analyzed according to the protocol recommended by Minekus and Alminger [22]. To prepare simulated salivary fluid (SSF), simulated gastric fluid (SGF) and simulated intestinal fluid (SIF), stock solutions of KCl, $KH_2PO_4$, $NaHCO_3$, NaCl, $MgCl_2(H_2O)_6$ and $(NH_4)_2CO_3$ were used and kept at 37 °C for use.

In the oral phase, 1 g of sample was added to 0.7 mL of SSF, 5 μL of 0.3 M $CaCl_2$ and 0.295 mL of deionized water in 50 mL falcon tubes and incubated with agitation for 2 min at 37 °C (TE-053, Tecnal, Piracicaba, Brazil). Subsequently, for the gastric phase, 1.5 mL of SGF and 1 μL of 0.3 M $CaCl_2$ were added to the tubes containing the oral phase, the pH was corrected to 3 using 6 M HCl and the tubes were filled with deionized water until to reach a final volume of 4 mL in each tube. Then, 0.32 mL of 25,000 U/mL pepsin solution was added, and after homogenization, the tubes were incubated at 37 °C for 120 min. After 120 min of incubation, for the intestinal phase, 2.2 mL of SIF was added to the tubes containing the gastric phase, and the liquids were homogenized; then, 0.5 mL of 160 mM bile solution and 8 μL of $CaCl_2$ 0.3 M were added. M. The pH was then corrected to 7 with 5 M NaOH solution, and deionized water was added until reaching a final volume of 8 mL in each test tube. Subsequently, 1 mL each of 800 U/mL pancreatin solution was added to the tubes, they were homogenized and incubated at 37 °C for 120 min.

For evaluation purposes, aliquots were removed at the end of the oral phase (2 min), the middle (60 min) and late (120 min) gastric phases (5 M NaOH solution was added

to each tube to reach pH 7 and cease pepsin activity) and the middle (60 min) and late (120 min) intestinal phases (6 M HCl solution was added to each to reach pH 5 and cease pancreatin activity). The tubes were centrifuged (Genesys 10S UV-Visible, Thermo Fisher Scientific, Waltham, MA, USA) at 7830 rpm for 5 min, and aliquots of the supernatant were removed for analysis of total phenolic compounds according to the methodology described in item 2.4.6.

### 2.7. Statistical Analysis

The results were evaluated by means of the analysis of variance (ANOVA), and when applicable, the Tukey mean test with a 95% confidence interval ($p < 0.05$), using the Statistica Software (Statistica, version 10, Statsoft. Inc., Tulsa, OK, USA).

### 3. Results

#### 3.1. Characterization of Powders

Powders were produced by spray drying with papaya seed extracts and maltodextrin as the carrier agent, which were characterized in terms of yield, moisture, water activity, hygroscopicity, dispersibility, color parameters (L*, a*, b*) and mean particle diameter, as shown in Table 2.

**Table 2.** Yield, moisture content, water activity, hygroscopicity, dispersibility, color parameters (L*, a*, b*) and average particle diameter of each powder produced by spray drying of papaya seed extracts with maltodextrin as a carrier.

| Fruit Ripening Stage | Maltodextrin Concentration | Drying Temperature | |
| --- | --- | --- | --- |
| | | **130 °C** | **150 °C** |
| Yield (%) | | | |
| Unripe | 0% | 71.6 ± 0.1 [A.a] | 71.7 ± 1.8 [A.a] |
| Unripe | 15% | 74.7 ± 0.3 [A.a] | 76.1 ± 1.3 [A.a] |
| Unripe | 30% | 71.7 ± 1.6 [A.a] | 71.1 ± 1.4 [A.a] |
| Ripe | 0% | 62.4 ± 0.1 [A.a] | 52.9 ± 10.4 [A.a] |
| Ripe | 15% | 67.4 ± 1.7 [A.a] | 66.6 ± 9.0 [A.a] |
| Ripe | 30% | 65.0 ± 4.7 [A.a] | 60.1 ± 3.5 [A.a] |
| Moisture content (%) | | | |
| Unripe | 0% | 4.2 ± 0.3 [A.a] | 3.6 ± 0.3 [A.a] |
| Unripe | 15% | 5.2 ± 0.2 [A.a] | 4.7 ± 1.1 [A.a] |
| Unripe | 30% | 4.0 ± 0.5 [A.a] | 3.6 ± 0.2 [A.a] |
| Ripe | 0% | 6.4 ± 0.6 [A.a] | 6.3 ± 1.7 [A.a] |
| Ripe | 15% | 4.5 ± 0.7 [A.a] | 4.2 ± 1.0 [A.a] |
| Ripe | 30% | 4.6 ± 0.4 [A.a] | 4.5 ± 0.5 [A.a] |
| Water activity (Aw) | | | |
| Unripe | 0% | 0.224 ± 0.0 [AB.a] | 0.182 ± 0.0 [B.a] |
| Unripe | 15% | 0.294 ± 0.0 [A.a] | 0.319 ± 0.0 [A.a] |
| Unripe | 30% | 0.175 ± 0.0 [B.a] | 0.208 ± 0.0 [A.a] |
| Ripe | 0% | 0.304 ± 0.01 [A.a] | 0.338 ± 0.1 [A.a] |
| Ripe | 15% | 0.222 ± 0.1 [A.a] | 0.194 ± 0.1 [A.a] |
| Ripe | 30% | 0.188 ± 0.0 [A.a] | 0.181 ± 0.0 [A.a] |
| Hygroscopicity (g of absorbed water/100 g of powder) | | | |
| Unripe | 0% | 25.0 ± 2.2 [A.a] | 25.0 ± 0.5 [A.a] |
| Unripe | 15% | 15.0 ± 2.4 [B.a] | 14.1 ± 1.0 [B.a] |
| Unripe | 30% | 12.1 ± 1.0 [B.a] | 12.8 ± 1.8 [B.a] |
| Ripe | 0% | 31.5 ± 0.9 [A.a] | 30.6 ± 1.6 [A.a] |
| Ripe | 15% | 14.5 ± 1.1 [B.a] | 15.2 ± 1.3 [B.a] |
| Ripe | 30% | 12.3 ± 0.4 [B.a] | 11.4 ± 1.5 [B.a] |

**Table 2.** *Cont.*

| Fruit Ripening Stage | Maltodextrin Concentration | Drying Temperature | |
|---|---|---|---|
| | | 130 °C | 150 °C |
| *Dispersibility (%)* | | | |
| Unripe | 0% | 82.4 ± 0.7 [B.b] | 86.0 ± 1.1 [B.a] |
| Unripe | 15% | 91.0 ± 0.3 [A.a] | 91.0 ± 1.9 [A.a] |
| Unripe | 30% | 93.0 ± 1.3 [A.a] | 93.4 ± 2.0 [A.a] |
| Ripe | 0% | 89.2 ± 0.9 [B.a] | 87.1 ± 0.9 [B.a] |
| Ripe | 15% | 94.2 ± 2.0 [A.a] | 94.5 ± 2.1 [A.a] |
| Ripe | 30% | 94.3 ± 0.8 [A.a] | 96.6 ± 0.4 [A.a] |
| *Color parameter L\** | | | |
| Unripe | 0% | 50.6 ± 1.1 [C.a] | 43.5 ± 1.1 [B.a] |
| Unripe | 15% | 66.0 ± 0.0 [B.a] | 68.7 ± 1.1 [A.a] |
| Unripe | 30% | 72.8 ± 1.0 [A.a] | 69.7 ± 1.1 [A.a] |
| Ripe | 0% | 33.3 ± 1.0 [B.a] | 30.8 ± 0.5 [C.a] |
| Ripe | 15% | 72.5 ± 1.7 [A.a] | 74.4 ± 0.8 [B.a] |
| Ripe | 30% | 75.3 ± 2.7 [A.a] | 77.1 ± 0.5 [A.a] |
| *Color parameter a\** | | | |
| Unripe | 0% | 6.7 ± 0.4 [A.b] | 9.5 ± 0.0 [A.a] |
| Unripe | 15% | 2.7 ± 0.2 [B.a] | 2.5 ± 0.0 [B.a] |
| Unripe | 30% | 2.1 ± 0.2 [B.a] | 2.4 ± 0.0 [B.a] |
| Ripe | 0% | 11.6 ± 0.1 [A.a] | 12.0 ± 0.0 [A.a] |
| Ripe | 15% | 2.3 ± 0.0 [B.a] | 2.3 ± 0.6 [B. a] |
| Ripe | 30% | 2.0 ± 0.5 [B.a] | 2.0 ± 0.0 [B.a] |
| *Color parameter b\** | | | |
| Unripe | 0% | 24.8 ± 0.8 [A.b] | 28.1 ± 0.5 [A.a] |
| Unripe | 15% | 14.9 ± 0.4 [B.a] | 16.3 ± 0.4 [B.a] |
| Unripe | 30% | 13.3 ± 0.7 [B.b] | 15.2 ± 1.0 [B.a] |
| Ripe | 0% | 27.2 ± 0.2 [A.a] | 21.7 ± 5.7 [A.a] |
| Ripe | 15% | 13.8 ± 0.2 [B.a] | 14.8 ± 1.4 [AB.a] |
| Ripe | 30% | 12.6 ± 1.0 [B.a] | 12.8 ± 0.1 [B.a] |
| *Volume-weighted average diameter (μm)* | | | |
| Unripe | 0% | 16.1 ± 0.3 [A.a] | 26.8 ± 2.7 [A.b] |
| Unripe | 15% | 6.6 ± 2.0 [B.a] | 6.4 ± 2.0 [B.a] |
| Unripe | 30% | 9.1 ± 2.5 [AB.a] | 11.7 ± 0.4 [B.a] |
| Ripe | 0% | 109.5 ± 8.1 [A.a] | 130.0 ± 5.7 [A.a] |
| Ripe | 15% | 11.0 ± 1.7 [B.a] | 7.0 ± 2.0 [B.a] |
| Ripe | 30% | 16.3 ± 2.1 [B.a] | 14.8 ± 3.2 [B.a] |

±Means followed by the standard deviations. Means in the same column followed by a capital letter do not differ from each other at the level of 5% ($p < 0.05$) probability by Tukey's test, and means in the same row followed by a lower-case letter do not differ from each other at the level of 5% ($p < 0.05$) probability by Tukey's test.

Based on the results, yield and moisture values were not affected by differences in maltodextrin concentrations (0%, 15%, and 30%), which was an unexpected result, since the presence of the carrier usually positively influences yield due to the higher solids content in the feed. The different temperatures (130 and 150 °C) of the inlet air also did not influence these parameters.

The unripe seed extract powders obtained higher yields than the ripe seed extract powders; however, the two types of extract showed good yield rates, all above 50%. For the water activity parameter, the powders of unripe seed extracts dried at 130 °C were influenced by the maltodextrin concentration, as the values were lower the higher the concentration of the drying aid in the feed. At the temperature of 150 °C, the increase in the concentration of the carrier agent did not clearly change the Aw values for the extract

obtained from the unripe fruit; for the extract obtained from the ripe fruit, the same effect was observed as in the drying at 130 °C.

The moisture and water activity values of the powders obtained can be considered low and are within the levels recommended for powders obtained by spray drying. Furthermore, the low values of Aw allow us to conclude that the powders obtained were stable from a microbiological point of view, since there is no possibility of microorganisms multiplying in materials whose water activity is lower than 0.6 [23].

It is observed that the presence of maltodextrin contributed significantly to the reduction of the hygroscopicity of the powders. Drying temperatures did not affect the hygroscopicity of the powders in the two types of papaya seed extracts. This result may be related to the fact that maltodextrin is less hygroscopic than the extract. The low hygroscopicity of maltodextrin confirms its viability and importance as a carrier in spray-drying processes [24]. Similar results were found by Calomeni and de Souza [12], for powders of peanut skin extracts obtained by spray drying, which observed that increasing maltodextrin concentration produced powders with low hygroscopicity.

For the dispersibility parameter, and for hygroscopicity, it was observed that the presence of maltodextrin increased the dispersibility of the powders obtained. Dry extracts without the addition of a carrier agent were less dispersible, thereby showing that maltodextrin contributes to improving this property because it is a polymer with high water dispersibility and has many hydroxyl groups [25]. Powders of unripe seed extracts dried at 150 °C showed higher dispersibility than powders dried at 130 °C. In the other concentrations and in the powders of extract of ripe seeds there were no significant changes. de Souza and Thomazini [26], when using maltodextrin as a carrier agent in the development of powdered pigments from grape by-products (peels and seeds) using the spray drying technique, observed that the carrier agent promoted greater dispersibility than the lyophilized powders (without carrier), reaching average dispersibility values around 96%, levels close to those found in the present study.

When evaluating the color parameter L* (luminosity) of the samples, it was observed that the samples produced without the carrier agent had lower values of luminosity, which tend to have a darker color when compared to those with the addition of maltodextrin. The increase in maltodextrin concentration promoted an increase in the luminosity of the samples, due to the dilution of the pigments present in the extract; therefore, the higher the carrier concentration, the greater the luminosity of the samples. The drying temperature did not affect the luminosity of the samples. The dry powders of unripe papaya seed extract without the addition of maltodextrin showed lower luminosity than the dry powders of ripe papaya seed extract, since they presented less intense original colors or a smaller number of substances capable of absorbing light and reflecting in the region of the spectrum that resulted in more intense colors.

It is observed that in relation to the parameter a*, which measures the intensity of color from unripe (−) to red (+), all samples have positive values, and indeed, they tended to have a red color. The addition of maltodextrin reduced the values of the red color intensity due to the dilution of the pigments originally present in the extract.

The appearance of the powders is shown in Figure 1. Maltodextrin caused a dilution of the pigments present in the extracts, contributing to a less intense red color. de Souza and Thomazini [26] also observed in their work that the dry extract of burgundy grape winemaking residues without a carrier agent showed a higher intensity of red color than samples with higher concentrations of maltodextrin.

The drying temperature had a significant influence only on the values of the color parameters a* and b* of the pure extracts obtained from seeds of unripe fruits. This result can be attributed to the high concentration of pigments in this sample.

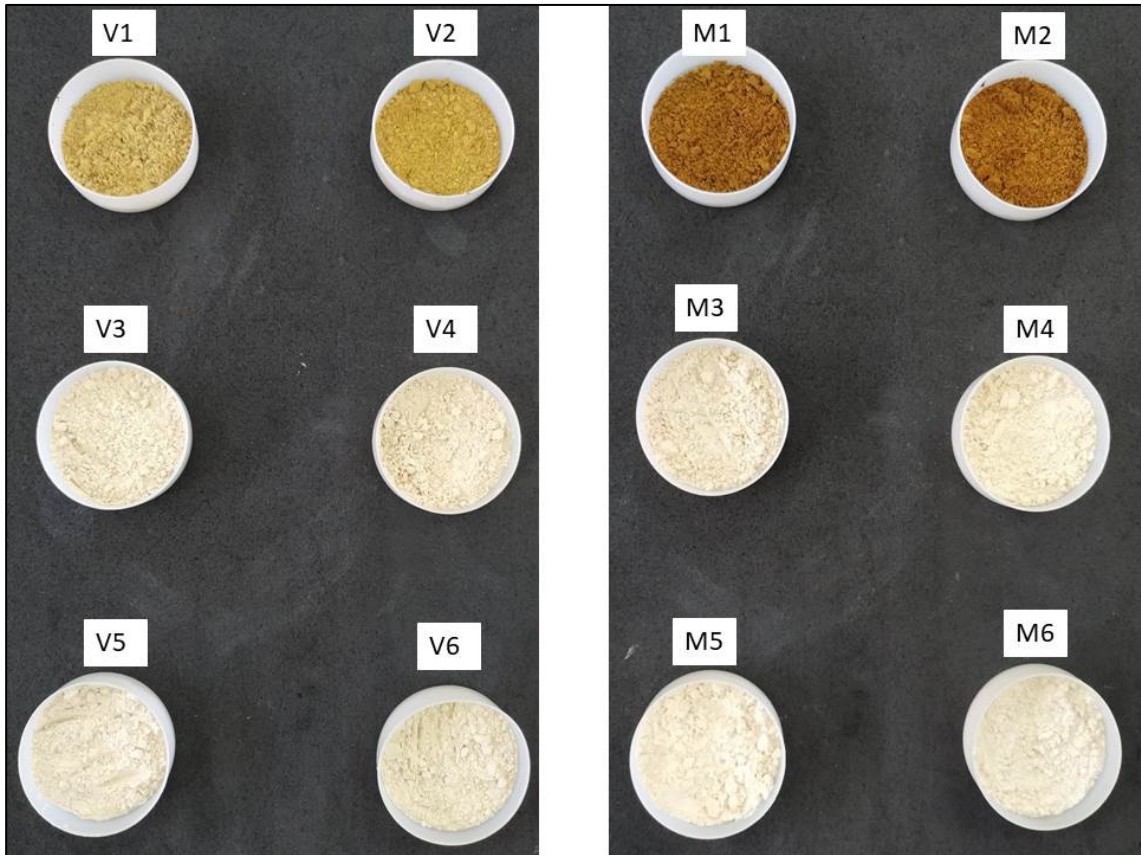

**Figure 1.** Appearance of powders obtained by spray drying of papaya seed extracts. In this figure, V1: unripe seed extract with 0% maltodextrin and dried at 130 °C; V2: unripe seed extract with 0% maltodextrin and dried at 150 °C; V3: unripe seed extract with 15% maltodextrin and dried at 130 °C; V4: unripe seed extract with 15% maltodextrin and dried at 150 °C; V5: unripe seed extract with 30% maltodextrin and dried at 130 °C; V6: unripe seed extract with 30% maltodextrin and dried at 150 °C; M1: mature seed extract with 0% maltodextrin and dried at 130 °C; M2: mature seed extract with 0% maltodextrin and dried at 150 °C; M3: mature seed extract with 15% maltodextrin and dried at 130 °C; M4: mature seed extract with 15% maltodextrin and dried at 150 °C; M5: mature seed extract with 30% maltodextrin and dried at 130 °C; M6: mature seed extract with 30% maltodextrin and dried at 150 °C.

Considering the empirical parameter of visual appearance (Figure 1), it can be clearly seen that samples M1 and M2, prepared from the seeds of ripe fruits were more pigmented and more agglomerated than the others. This was corroborated by the scanning electron micrographs (Figure 2) and by the result of the average particle sizes (Table 2). These samples originally had more pigments, possibly suffered more browning reactions during drying and agglomerated more, as they were more hygroscopic. It is also clear that maltodextrin dilutes the pigments present in the extracts and must also inhibit chemical browning reactions during drying process. However, it is not clear that the increase from 15 to 30% of maltodextrin in the formulation was effective for that purposes.

Regarding the parameter b*, which measures the intensity of blue (−) to yellow (+) colors, all samples presented positive values, which is in accordance with their appearance. The increase in maltodextrin concentration caused a reduction in the intensity of yellow coloration in the powders obtained. The increase in drying temperature caused an increase in the intensity of the yellow color parameter in the powder samples of unripe seed extracts with 0% and 30% maltodextrin. This tendency was not very clear, since it was not noticed in samples with 15% maltodextrin and in those ones produced with ripe seeds; however, it can be attributed to a slight intensification of browning reactions.

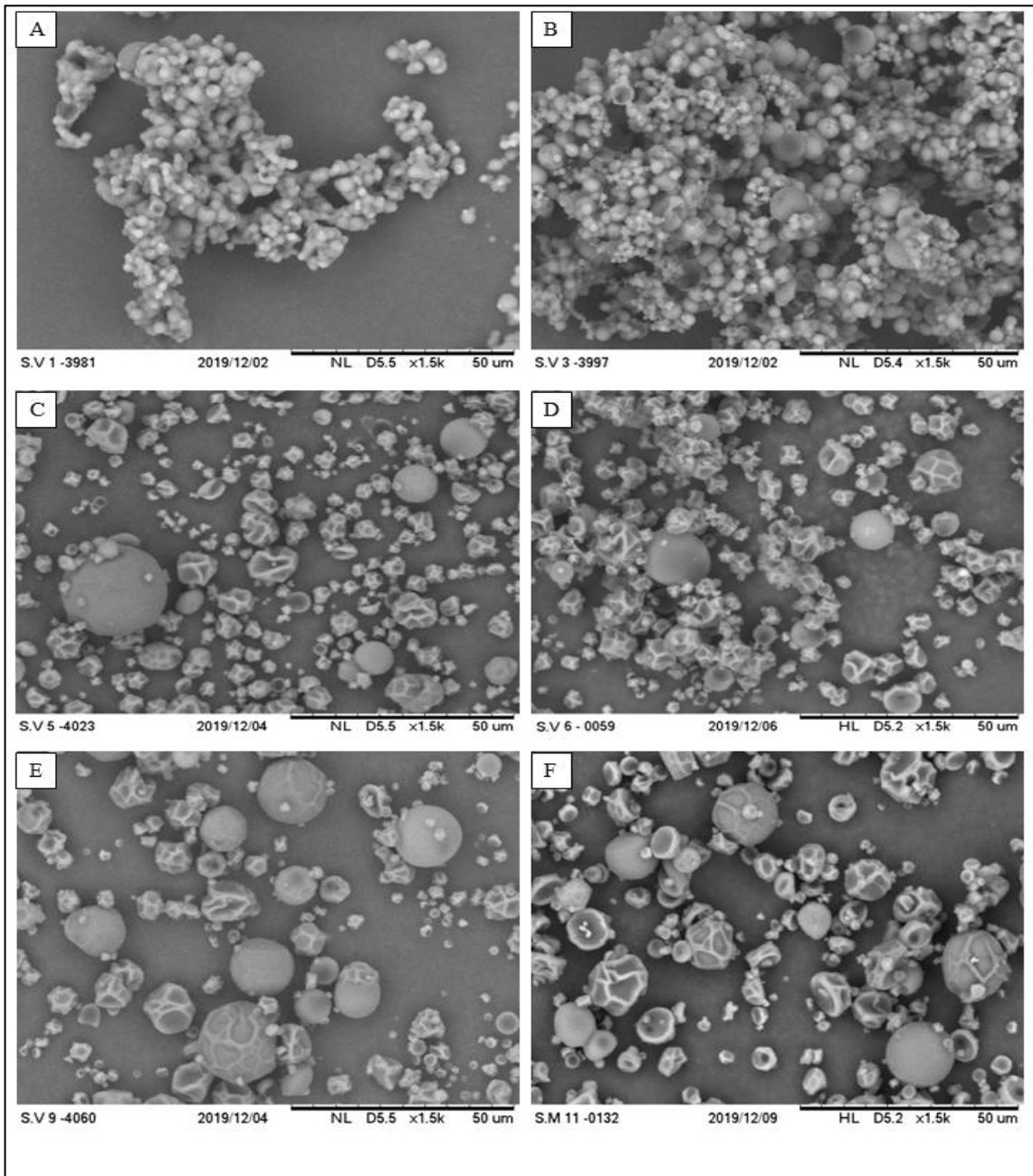

**Figure 2.** Micrographs of particles of ripe papaya seed extracts sprayed at 130 and 150 °C. In this figure: (**A**) sample with 0% maltodextrin and dried at 130 °C; (**B**) sample with 0% maltodextrin and dried at 150 °C; (**C**): sample with 15% maltodextrin and dried at 130 °C; (**D**) sample with 15% maltodextrin dried at 150 °C; (**E**) sample with 30% maltodextrin increased and dried at 130 °C; (**F**) sample with 30% maltodextrin and dried at 150 °C.

The size of the particles is an important parameter in defining their applications, since particles larger than 100 µm can have negative influences on the sensory acceptance of products. It was observed that the dried samples without the addition of the carrier agent had larger particle diameters; this may be related to the agglomeration of this particles (Figure 2), due to their higher hygroscopicity.

For both extracts, the hygroscopicity was higher in the samples without the addition of maltodextrin; this greater hygroscopicity may have increased the agglomeration capacity of these samples, as can also be observed with the morphology analysis, forming particles with larger diameters. The dry extract of unripe papaya seeds (free) presented smaller

diameter (16.1 μm dry at 130 °C and 26.8 μm dry at 150 °C) in relation to the dry extract (free) of seeds of ripe papaya (109.05 μm dry at 130 °C and 130 μm dry at 150 °C). Again, these marked differences were probably caused by more intense particle agglomeration of ripe seeds' dry extract, not only due to its higher hygroscopicity, but also higher moisture and Aw when compared to the unripe seeds' powder.

The average diameter of samples of ripe seeds without the addition of maltodextrin was greater than 50 μm, which would have a negative influence on the texture of food, thereby demonstrating the importance of using the carrier agent, since samples with maltodextrin had average diameters lower than 50 μm. The drying temperature significantly influenced only the dry extract of unripe papaya seeds without the addition of maltodextrin, which can be explained by the temperature of 150 °C causing more caramelization, as can be seen in the color analysis, promoting greater agglomeration of the particles. de Souza and Fujita [12] observed that the increase in temperature did not affect the particle diameters of powder samples of grape by-products obtained by spray drying and added with maltodextrin.

### 3.1.1. Particle Morphology

The micrographs of the powders produced in this study were obtained by scanning electron microscopy (SEM) and are represented in Figure 2.

It is observed that the microparticles, in general, had variable diameters, with agglomerations, a spherical shape and rough surfaces. It is important to obtain spherical particles for easier powder application and better flow properties [27].

The dried samples without the addition of maltodextrin showed greater agglomeration in both extracts when compared to the samples with the carrier agent, and the increase in temperature—extracts were dried at 150 °C in the free form—caused a greater diameter and greater agglomeration. The dry extract of ripe seeds (free) presented greater agglomeration of the particles in relation to the dry extract of unripe seeds (free), which can be explained by the fact that these samples present greater hygroscopicity, facilitating greater absorption of water and thus agglomeration of the particles.

The increase in maltodextrin concentration did not visually affect the morphology of the particles. However, because the extract and maltodextrin formed a solution in the feed, they also formed the walls of the particles, which were possibly hollow. The particles thus constituted were of the matrix type and can also be called microspheres, but not microcapsules.

### 3.1.2. Total Phenolic Content and Antioxidant Activity of Powders

Powders obtained by spray drying with papaya seed extracts and maltodextrin as carrier agents were also characterized in terms of total phenolic content and antioxidant activity, as shown in Table 3. The dry extract of unripe papaya seed had 63.5 mg of EAG/g of powder (130 °C) and 71.55 mg of EAG/g of powder (150 °C), and for the dry extract of ripe seeds, 44 mg of EAG/g of powder (130 °C) and 44.2 mg of EAG/g of powder (150 °C). In both extracts, the drying temperature did not significantly influence the phenolic content; on the other hand, the increase in maltodextrin concentration caused a reduction in these levels, which is related to the concentration of the carrier agent—that is, maltodextrin caused the dilution of the compounds originally present in the extract via antioxidant action. This reduction was also observed in the antioxidant activity by the ABTS and FRAP methods.

The dry extract of unripe seeds had a higher phenolic content than the extract of ripe seeds, which can be explained by the ripening of the fruit, during which the degradation of phenolics and/or their polymerization can occur. Degradation implies loss, and polymerization can change dispersibility, directly affecting extraction efficiency. It is known that phenolic compounds have antioxidant action. In this work, it was observed that the powders with higher levels of these compounds had higher levels of antioxidant activity, as evidenced by the results obtained for the ABTS and FRAP methods. However, it should be noted that the antioxidant activity was different for each compound and depends on other factors, such as dispersibility in the medium and affinity with the free radical, so

some molecules are more potent than others; thus, it is not possible to categorically state for all extracts that the antioxidant activity is always proportional to the concentration of phenolics, as occurred in this work for the extract obtained from the seeds of unripe fruits.

**Table 3.** Total phenolic content and antioxidant activity of spray-dried powders of pure papaya seed extracts and those with maltodextrin as a carrier agent.

| Fruit Ripening Stage | Maltodextrin Concentration | Drying Temperature | |
|---|---|---|---|
| | | 130 °C | 150 °C |
| | *Total phenolics (mg of gallic acid/g of powder)* | | |
| Unripe | 0% | 63.5 ± 0.1 [A,a] | 71.55 ± 0.1 [A,a] |
| Unripe | 15% | 16.6 ± 1.0 [B,a] | 16.4 ± 0.1 [B,a] |
| Unripe | 30% | 8.9 ± 0.3 [C,a] | 10.3 ± 0.5 [C,a] |
| Ripe | 0% | 44.0 ± 0.1 [A,a] | 44.2 ± 0.5 [A,a] |
| Ripe | 15% | 5.5 ± 0.1 [B,a] | 5.4 ± 0.1 [B,a] |
| Ripe | 30% | 3.2 ± 0.1 [C,a] | 3.1 ± 0.1 [C,a] |
| | *ABTS$^{\bullet+}$ (µM of Trolox/g of powder)* | | |
| Unripe | 0% | 735.95 ± 4.7 [A,b] | 783.81 ± 10.7 [A,a] |
| Unripe | 15% | 102.4 ± 2.0 [B,a] | 106.02 ± 5.7 [B,a] |
| Unripe | 30% | 76.4 ± 2.3 [C,a] | 81.45 ± 0.5 [C,a] |
| Ripe | 0% | 442.5 ± 3.7 [A,b] | 486.7 ± 0.1 [A,a] |
| Ripe | 15% | 71.2 ± 2.0 [B,a] | 70.5 ± 1.3 [B,a] |
| Ripe | 30% | 44. 1 ± 0.1 [C,a] | 43.8 ± 0.0 [C,a] |
| | *FRAP (µM of ferrous sulfate/g of powder)* | | |
| Unripe | 0% | 269.5 ± 1.0 [A,b] | 292.2 ± 6.9 [A,a] |
| Unripe | 15% | 64.3 ± 3.8 [B,a] | 63.1 ± 0.6 [B,a] |
| Unripe | 30% | 37.9 ± 0.1 [C,a] | 38.7 ± 0.4 [C,a] |
| Ripe | 0% | 360.6 ± 5.2 [A,b] | 388.7 ± 0.8 [A,a] |
| Ripe | 15% | 53.5 ± 0.9 [B,a] | 53.7 ± 1.1 [B,a] |
| Ripe | 30% | 27.8 ± 0.3 [C,a] | 28.2 ± 1.0 [C,a] |

±Means followed by the standard deviations. Means in the same column followed by a capital letter do not differ from each other at the level of 5% ($p < 0.05$) probability by Tukey's test, and means in the same row followed by a lower-case letter do not differ from each other at the level of 5% ($p < 0.05$) probability by Tukey's test.

However, even though a lower content of total phenolics was detected in the extract powder obtained from seeds of ripe fruit, the antioxidant activity by the FRAP method was higher when compared to the powder of the extract obtained from seeds of unripe fruit, indicating that relevant compounds remained by the reduction of the ferric complex ($Fe^{3+}$). The dry extracts of papaya seeds showed antioxidant activity by the ABTS method superior to that of other fruits, such as guava (15.31 µM Trolox/g pulp) [28], passion fruit and mango (25 and 18.7 µM of Trolox/g pulp (dry basis)) [29], even at the highest concentration of maltodextrin. However, as far as is known, this is the first study that performed the encapsulation of papaya seed extracts by the spray-drying technique, which makes comparison with literature data impossible.

### 3.1.3. Color Stability and Phenolic Content of Powders during Storage

The total color difference (ΔE*) of the samples was evaluated at the end of storage (90 days) at 25 °C and 32.8% relative humidity. The results presented in Table 4 indicate that the addition of maltodextrin protected the pigments, since ΔE values were smaller for powders dried with this carrier. Therefore, there was an encapsulating effect, promoting the preservation of the color of the powders. The drying temperature for the unripe seed extract did not affect the total color loss of the powders. Similar results were found by Calomeni and de Souza [13] when evaluating the influence of the carrier agent on the color stability of dried peanut extracts by spray drying, and the maltodextrin protected the pigments when compared to lyophilized samples.

**Table 4.** Color stability of spray dried powders with papaya seed extracts and maltodextrin as a carrier.

| Fruit Ripening Stage | Maltodextrin Concentration | Drying Temperature | |
|---|---|---|---|
| | | **130 °C** | **150 °C** |
| | | Total color difference ($\Delta E^*$) | |
| Unripe | 0% | 47.6 ± 0.1 [A,a] | 41.4 ± 4.0 [A,a] |
| Unripe | 15% | 17.1 ± 1.3 [B,a] | 21.7 ± 1.7 [B,a] |
| Unripe | 30% | 22.1 ± 1.9 [B,a] | 21.6 ± 0.6 [B,a] |
| Ripe | 0% | 29.0 ± 2.0 [B,a] | 18.0 ± 2.8 [B,b] |
| Ripe | 15% | 20.8 ± 2.9 [A,a] | 22.0 ± 0.6 [AB,a] |
| Ripe | 30% | 21.8 ± 3.4 [A,a] | 25.0 ± 0.5 [A,a] |

±Means followed by the standard deviation. Means in the same column followed by a capital letter do not differ from each other at the level of 5% ($p < 0.05$) probability by Tukey's test, and means in the same row followed by a lower case letter do not differ from each other at the level of 5% ($p < 0.05$) probability by Tukey's test.

For the dry extract of ripe papaya seeds, it was observed that the drying temperature had an influence only on the extracts without the addition of maltodextrin, which dried at 130 °C and had greater loss. Regarding the concentration of carrier agent, the extracts dried at 150 °C had greater loss as the concentration increased.

Another factor observed in the samples without the addition of maltodextrin in both extracts was the occurrence of the caking phenomenon. The samples suffered darkening, wrinkling, caking, and particle agglomeration after 90 days of storage, as can be seen in Figure 3.

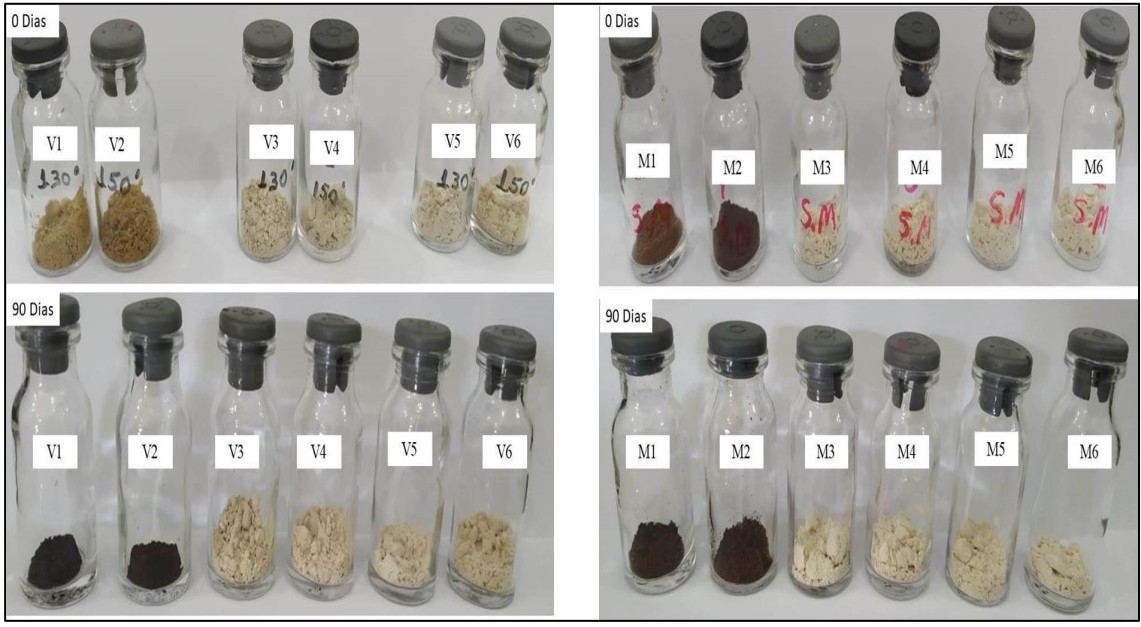

**Figure 3.** Appearances of powders in the very beginning (**top**) and after 90 days of storage (**bottom**). V1, V3 and V5: Unripe seed extracts with 0%, 15% and 30% maltodextrin, respectively, dried at 130 °C; V2, V4 and V6: Unripe seed extracts with 0%, 15% and 30% maltodextrin, respectively, dried at 150 °C; M1, M3 and M5: Ripe seed extracts with 0%, 15% and 30% maltodextrin, respectively, dried at 130 °C; M2, M4 and M6: Ripe seed extracts with 0%, 15% and 30% maltodextrin, respectively, dried at 150 °C.

Caking consists of changes in food powders due to changes in the environment, such as a switch in humidity that induces plasticization on the surface of the particle, changing it into a hardened and agglomerated material, causing loss of functionality and quality, and thereby, economic losses [30,31]. The dry extracts with the addition of the carrier agent

did not undergo caking. This can be explained due to the hygroscopicity of the powders, which is associated with the ability to absorb moisture from the environment. The powders of the extracts without the addition of maltodextrin showed higher levels of hygroscopicity. Therefore, they absorb more water, facilitating degradation and suffering caking.

The total phenolic contents of the samples were evaluated every 15 days during storage for 90 days at 25 °C and 38.2% relative humidity, and the results are shown in Table 5.

**Table 5.** Stability of total phenolics of spray-dried powders from papaya seed extracts with maltodextrin as a carrier.

| Total Phenolics (mg GAE/g Powder) | | | | |
|---|---|---|---|---|
| | | | **Drying Temperature** | |
| **Fruit Ripening Stage** | **Maltodextrin Concentration (%)** | **Days** | **130 °C** | **150 °C** |
| Unripe | 0 | 0 | $63.5 \pm 0.2$ [A,b] | $71.5 \pm 0.2$ [A,a] |
| | | 15 | $57.8 \pm 0.5$ [B,b] | $64.3 \pm 1.8$ [B,a] |
| | | 30 | $57.9 \pm 0.8$ [B,a] | $64.3 \pm 2.3$ [B,a] |
| | | 45 | $57.8 \pm 0.5$ [B,a] | $60.9 \pm 1.7$ [BC,a] |
| | | 60 | $54.0 \pm 1.3$ [C,a] | $57.1 \pm 2.5$ [C,a] |
| | | 75 | $52.0 \pm 0.1$ [C,b] | $64,4 \pm 0.5$ [B,a] |
| | | 90 | $52.6 \pm 0.1$ [C,a] | $58.3 \pm 2.0$ [BC,a] |
| Unripe | 15 | 0 | $16.6 \pm 1.5$ [A,a] | $16.4 \pm 0.2$ [A,a] |
| | | 15 | $15.0 \pm 0.3$ [AB,a] | $16.3 \pm 1.1$ [A,a] |
| | | 30 | $15.0 \pm 0.1$ [AB,a] | $15,2 \pm 0.8$ [A,a] |
| | | 45 | $14.4 \pm 0.2$ [AB,a] | $15.1 \pm 1.0$ [A,a] |
| | | 60 | $14.4 \pm 0.0$ [AB,a] | $14.8 \pm 0.3$ [A,a] |
| | | 75 | $13.7 \pm 0.7$ [B,a] | $14.9 \pm 0.4$ [A,a] |
| | | 90 | $15.24 \pm 0.1$ [AB,a] | $14.9 \pm 0.5$ [A,a] |
| Unripe | 30 | 0 | $9.0 \pm 0.4$ [A,a] | $10.3 \pm 0.7$ [A,a] |
| | | 15 | $8.5 \pm 0.3$ [A,a] | $10.2 \pm 2.0$ [A,a] |
| | | 30 | $8.3 \pm 0.3$ [A,a] | $10.1 \pm 0.5$ [A,b] |
| | | 45 | $8.3 \pm 0.2$ [A,a] | $9.8 \pm 0.8$ [A,a] |
| | | 60 | $8.5 \pm 0.1$ [A,a] | $9.8 \pm 1.4$ [A,a] |
| | | 75 | $8.6 \pm 0.4$ [A,a] | $10.2 \pm 0.1$ [A,b] |
| | | 90 | $8.8 \pm 0.3$ [A,a] | $9.4 \pm 0.9$ [A,a] |
| Ripe | 0 | 0 | $44.0 \pm 0.8$ [A,a] | $44.2 \pm 1.4$ [A,a] |
| | | 15 | $43.3 \pm 1.4$ [A,a] | $43.6 \pm 4.7$ [A,a] |
| | | 30 | $41.8 \pm 1.2$ [A,b] | $51.0 \pm 0.4$ [A,a] |
| | | 45 | $43.3 \pm 0.3$ [A,a] | $49.7 \pm 2.6$ [A,a] |
| | | 60 | $41.9 \pm 1.2$ [A,a] | $50.3 \pm 2.6$ [A,a] |
| | | 75 | $43.0 \pm 3.9$ [A,a] | $47.1 \pm 2.6$ [A,a] |
| | | 90 | $46.3 \pm 4.7$ [A,a] | $47.1 \pm 3.2$ [A,a] |
| Ripe | 15 | 0 | $5.5 \pm 0.2$ [A,a] | $5.5 \pm 0.1$ [A,a] |
| | | 15 | $5.3 \pm 0.4$ [A,a] | $5.4 \pm 0.1$ [A,a] |
| | | 30 | $4.9 \pm 0.4$ [A,a] | $5.3 \pm 0.1$ [A,a] |
| | | 45 | $5.0 \pm 0.6$ [A,a] | $5.5 \pm 0.2$ [A,a] |
| | | 60 | $4.8 \pm 0.5$ [A,a] | $5.2 \pm 0.1$ [A,a] |
| | | 75 | $4.7 \pm 0.5$ [A,a] | $5.1 \pm 0.4$ [A,a] |
| | | 90 | $4.8 \pm 0.6$ [A,a] | $5.5 \pm 0.2$ [A,a] |
| Ripe | 30 | 0 | $3.2 \pm 0.1$ [A,a] | $3.3 \pm 0.1$ [A,a] |
| | | 15 | $3.0 \pm 0.1$ [A,a] | $3.0 \pm 0.0$ [A,a] |
| | | 30 | $3.0 \pm 0.1$ [A,a] | $3.0 \pm 0.1$ [A,a] |
| | | 45 | $3.0 \pm 0.2$ [A,a] | $3.0 \pm 0.2$ [A,a] |
| | | 60 | $2.9 \pm 0.1$ [A,a] | $3.0 \pm 0.1$ [A,a] |
| | | 75 | $2.9 \pm 0.1$ [A,a] | $2.9 \pm 0.1$ [A,a] |
| | | 90 | $3.0 \pm 0.1$ [A,a] | $3.0 \pm 0.0$ [A,a] |

±Means followed by the standard deviation. Means in the same column followed by a capital letter do not differ from each other at the level of 5% ($p < 0.05$) probability by Tukey's test, and means in the same row followed by a lower-case letter do not differ from each other at the level of 5% ($p < 0.05$) probability by Tukey's test.

For the dried extract of unripe papaya seed, it was observed that the atomized samples without maltodextrin suffered losses concerning the total phenolic content during the entire

storage period. The drying temperature of 150 °C caused an increase in the phenolic content on the days 0, 15 and 75. These increases may be associated with the depolymerization of phenolic compounds and the fact that the Folin-Ciocalteu reagent suffers interference from some reducing substances, such as reducing sugars [32], resulting in higher contents of these compounds. The sample with the addition of 15% maltodextrin suffered slight degradation of phenolics over time. Similar results were found by Calomeni and de Souza [13] when evaluating the dry extract of peanut skin by spray drying, who observed an increase in the phenolic content as the drying temperature increased, indicating that the high temperature caused the depolymerization of the compounds.

For the dried extract of ripe papaya seeds, all powders with the addition of the carrier agent remained stable throughout the storage period. The sample without maltodextrin was influenced by the drying temperature. The powder dried at 150 °C showed higher phenolic content within 30 days.

Maltodextrin is a product resulting from the partial hydrolysis of starch, formed mainly by macromolecules. Thus, the macromolecules that make up maltodextrins provide physical protection for smaller molecules, and the protective effect increases with increasing concentration. In the extracts of unripe and ripe papaya seeds, maltodextrin had a protective effect on powders dried by spray drying during the storage period, evidencing the importance of using the carrier agent in the quality and stability of the powders. For the dry extract of ripe papaya seeds, all powders with the addition of the carrier agent remained stable throughout the storage period.

### 3.1.4. In Vitro Release of Phenolic Compounds

Based on the results of the previous analyses, the following was taken into account for selection of the best treatments for digestion assay: (1) adding maltodextrin to papaya seed extracts improved their physicochemical properties and chemical stability, but increasing the carrier amount from 15% to 30% had little impact on these changes; (2) the drying temperature had little effect on most evaluated parameters, so a lower temperature is more interesting for reducing costs and energy consumption; (3) bioactive release from particles containing unripe and ripe papaya seed extracts has not been compared before, and the release profile may be distinct due to their potentially different phenolic composition. Therefore, V3 (15% maltodextrin and dried at 130 °C) and M3 (15% maltodextrin and dried at 130 °C) were selected as the best treatments to evaluate the release of phenolic compounds in each phase of the in vitro digestion process. Results for simulated oral (SSF), gastric (SGF) and intestinal (SIF) phases of digestion are shown in Figure 4.

In this figure, V3 denotes 15% dry maltodextrin at 130 °C, and M3 denotes 15% dry maltodextrin at 130 °C. The bars identified by the same capital letters of the same treatments do not differ statistically. The bars identified by the same lowercase letters in the same time interval do not differ statistically ($p > 0.05$) by the Tukey means test.

Based on Figure 4, it is observed that for both formulations, the phenolics were mostly released in the oral phase (SSF). This result can be attributed to the high hydrophilicity of maltodextrin.

In the M3 formulation, there was no release in the gastric phase, so the phenolic content was lower than that released in the oral phase. Due to the reduced level of phenolics released at the end of the gastric phase, it is believed that the degradation of the compounds released in the oral and gastric phases may have occurred, since phenolic compounds are sensitive to acidic conditions, such as the presence of gastric juice.

Subsequently, the release occurred in the intestinal phase. It is suggested that one or both of the following possibilities occurred. First, there must have been a compound that during digestion under the effects of enzymatic interactions and pH [33] may have caused the release of phenolic acids at this stage even though there was no release in the gastric phase. Another possibility is the instability of phenolic compounds under alkaline conditions [34] caused depolymerization, thereby generating smaller molecules that were

more reactive with the Folin–Ciocalteu reagent, thereby interfering with the quantification of phenolics.

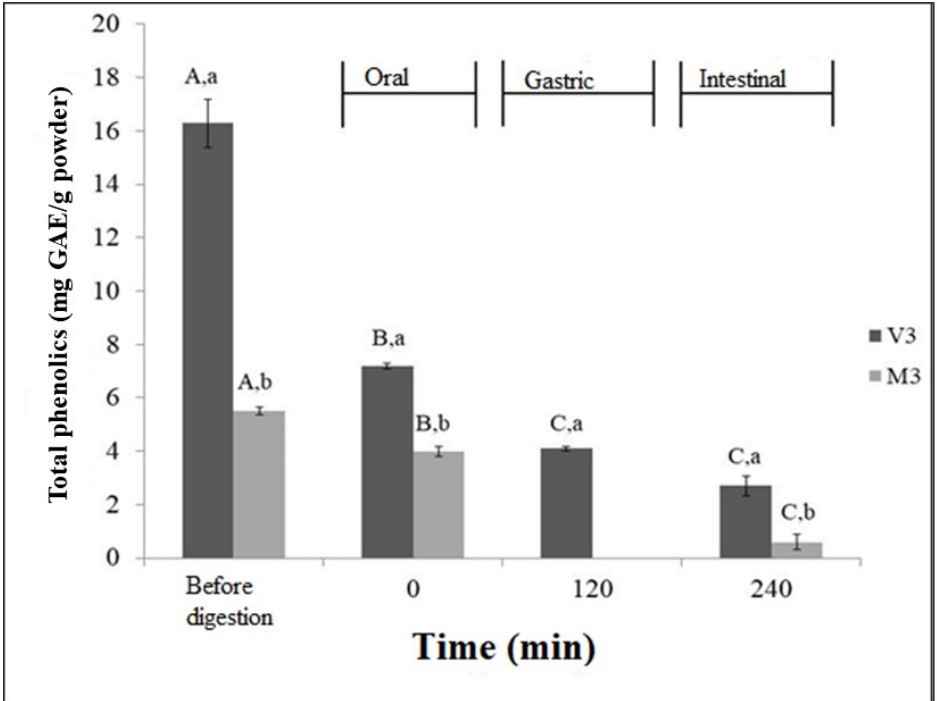

**Figure 4.** Release of phenolic compounds from encapsulated papaya seed extracts in simulated gastrointestinal fluids.

For both formulations, it was observed that the phenolic compounds did not reach total release and the intestinal phase caused a lower release of these compounds. This can be explained by the fact that the carrier agent reduced the capacity of release in this step, or the total release occurred, but during digestion, its degradation occurred due to extreme conditions of pH, enzymes, interaction with proteins and instability in alkaline conditions, as previously explained [33,34]. Therefore, further studies are necessary in order to identify and quantify phenolic compounds in isolation, thereby reducing interference in the quantification of these compounds.

## 4. Conclusions

Based on the results found in the present study, the addition of maltodextrin improves the quality of the powders obtained, providing increased dispersibility; and reducing hygroscopicity, agglomeration and mean particle diameter, in addition to the protective effect on the compounds during drying and storage period. Maltodextrin also protects the particles from caking and losses in the total phenolic content, thereby showing the importance of using the carrier agent for the quality and stability of the powders. Powders of extracts from unripe seeds showed higher levels of phenolic compounds and antioxidant activity compared to ripe seeds. Among the process conditions evaluated, treatments V3 and M3 had the best characteristics. For both treatments, the total phenolic content was released mostly in the simulated oral phase. Thus, the encapsulation of papaya seed extracts using maltodextrin as a carrier agent has technological potential, allowing its application to increase food quality and as a natural additive, in addition to being another viable alternative for the use of seeds from papaya processing, contributing to reducing the disposal of these wastes in the environment.

**Author Contributions:** Conceptualization, C.S.F.-T. and M.d.S.M.; methodology, C.S.F.-T., M.d.S.M. and M.T.; software, M.d.S.M. and M.T.; validation, M.d.S.M., M.T. and C.S.F.-T.; formal analysis, M.d.S.M. and M.T.; investigation, M.d.S.M., P.D.d.F.S. and A.T.H.; resources, M.T. and C.S.F.-T.; data curation, M.d.S.M. and M.T.; writing—original draft preparation, M.d.S.M., P.D.d.F.S. and A.T.H.; writing—review and editing, M.d.S.M., P.D.d.F.S., A.T.H., M.T. and C.S.F.-T.; visualization, M.d.S.M., P.D.d.F.S. and A.T.H.; supervision, C.S.F.-T.; project administration, C.S.F.-T. and M.d.S.M.; funding acquisition, not applicable. All authors have read and agreed to the published version of the manuscript.

**Funding:** This research received no external funding.

**Institutional Review Board Statement:** Not applicable.

**Informed Consent Statement:** Not applicable.

**Data Availability Statement:** Data available on request from the authors.

**Acknowledgments:** Authors thank Coordenação de Aperfeiçoamento de Pessoal de Nível Superior—Brazil (CAPES)—Finance Code 001. Favaro-Trindade also thanks Conselho Nacional de Desenvolvimento Científico e Tecnológico (CNPq) for the productivity grant (Process #305115/2018-9).

**Conflicts of Interest:** The authors declare that they have no conflict of interest.

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
