# Peer review of "Encapsulation of Formosa Papaya (Carica papaya L.) Seed Extract: Physicochemical Characteristics of Particles, and Study of Stability and Release of Encapsulated Phenolic Compounds"

_processes, doi:10.3390/pr11010027_

Round 1

Reviewer 1 Report

This article regards the interesting topic of circular economy, yet there are some changes to be done before publishing:

1)  The authors should inform the reader regarding the market of papaya, i.e. annual production, economic worth of the market as well as conventional ways of treating the wastes;

2) A more detailed chemical composition of seed extracts is required since, given the extraction protocols, it is almost certain that there are other molecular classes other than phenols, and this information is critical for any kind of potential commercialization.

Author Response

Reviewer 1: This article regards the interesting topic of circular economy, yet there are some changes to be done before publishing:

1)  The authors should inform the reader regarding the market of papaya, i.e. annual production, economic worth of the market as well as conventional ways of treating the wastes;

Answer: Thank you for your suggestion. The information was added and highlighted in yellow in the revised manuscript.

2) A more detailed chemical composition of seed extracts is required since, given the extraction protocols, it is almost certain that there are other molecular classes other than phenols, and this information is critical for any kind of potential commercialization.

Answer: The chemical composition of the extracts was treated in another study that has already been submitted and accepted by Food Science and Technology.

MESQUITA, Mércia da Silva et al. Papaya seeds (|Carica papayaL. var. Formosa) in different ripening stages: unexplored agro-industrial residues as potential sources of proteins, fibers and oil as well as high antioxidant capacity. Food Science and Technology [online]. 2023, v. 43 [Accessed 6 December 2022], e105422. Available from: <https://doi.org/10.1590/fst.105422>. Epub 05 Dec 2022. ISSN 1678-457X. https://doi.org/10.1590/fst.105422.

Reviewer 2 Report

In this manuscript, authors encapsulated papaya seed extract by spray drying at different concentration (0, 15, and 30%) of maltodextrin and different air-inlet temperature (130 and 150 OC). Authors conducted detailed analysis on the mcirocapsules and studied the release of bioactives from encapsulated powders. The manuscript was written well with detailed analysis. Following are my queries and suggestion:

The authors extracted the papaya seed from ripe and unripe seeds. What is the purpose of studying both unripe and ripe extracts. From table 3, encapsules of unripe seeds were showing maximum phenolic content and antioxidant activity than the ripe seeds. Why authors didn’t studied the phenolic content and antioxidant activity of extract alone (ripe and unripe) and choose the extract (that provided maximum antioxidant level) for encapsulation?

What was the outer temperature of the spray drying process? Similar to inlet temperature, outlet temperature also significantly affects the product characteristics.

Line no. 313 to 322, and line no. 325 to 328 were the captions for figure 1 and figure 2, respectively, but in text format. And the same paragraph was repeated in line 442-451.   

Author can explain why there is a huge deviation in particle size for unripe (0%) and ripe (0% maltodextrin) microcapsules. It will be interesting for the readers.

Line 492: Authors selected V3 and M3 for the digestion study and considered those were the best treatments. However, author didn’t reveal any scientific method conducted for considering best treatment. 

Author Response

Reviewer 2: In this manuscript, authors encapsulated papaya seed extract by spray drying at different concentration (0, 15, and 30%) of maltodextrin and different air-inlet temperature (130 and 150 OC). Authors conducted detailed analysis on the mcirocapsules and studied the release of bioactives from encapsulated powders. The manuscript was written well with detailed analysis. Following are my queries and suggestion:

The authors extracted the papaya seed from ripe and unripe seeds. What is the purpose of studying both unripe and ripe extracts. From table 3, encapsules of unripe seeds were showing maximum phenolic content and antioxidant activity than the ripe seeds. Why authors didn’t study the phenolic content and antioxidant activity of extract alone (ripe and unripe) and choose the extract (that provided maximum antioxidant level) for encapsulation?

Answer: Thank you for the comment.

At least in Brazil, both ripe and unripe papaya are processed for production of many products. So, the seeds from both stages of papayas maturation are waste here and certainly in many other countries. Besides, we expected differences in amount of phenolic compounds, for instance, considering the stage of maturation, and, indeed this was confirmed.

In fact, the authors studied and published the phenolic content and antioxidant activity of papaya seeds extracts in a previous work (https://doi.org/10.1590/fst.105422, mentioned in the comment above). Even though unripe seeds extract showed higher phenolic content and antioxidant activity than ripe seeds extract, the values found for ripe samples were still high. Also, the two types of seeds generate a significant amount of waste from the food industry, so in both cases it is important to find an alternative use for the materials. Finally, microencapsulation of unripe and ripe papaya seeds extracts had not been performed and compared before, especially regarding the release of encapsulated phenolics under simulated digestion, and the authors believe that publishing these results can contribute for future works in the field.

What was the outer temperature of the spray drying process? Similar to inlet temperature, outlet temperature also significantly affects the product characteristics.

Answer: The effect of the outlet air temperature on product characteristics was not evaluated in this work, and this parameter could not be controlled in the spray dryer equipment in which the experiments were performed, so it depended on the selected inlet air temperature. In a previous work of our research group, in which the same equipment was used, outlet air temperatures were recorded as 91.50-93.50 ºC for an inlet air temperature of 130 ºC; and 106.50-117.00 ºC for an inlet air temperature of 150 ºC (http://dx.doi.org/10.1016/j.fbp.2013.11.001).

Line no. 313 to 322, and line no. 325 to 328 were the captions for figure 1 and figure 2, respectively, but in text format. And the same paragraph was repeated in line 442-451. 

Answer: The corrections were implemented.

Author can explain why there is a huge deviation in particle size for unripe (0%) and ripe (0% maltodextrin) microcapsules. It will be interesting for the readers.

Answer: Authors thank the suggestion. The following sentence was added to the text:

Lines 363-365: “Again, these marked differences were probably caused by more intense particle agglomeration of ripe seeds’ dry extract, not only due to its higher hygroscopicity, but also higher moisture and Aw when compared to the unripe seeds’ powder.”

Line 492: Authors selected V3 and M3 for the digestion study and considered those were the best treatments. However, author didn’t reveal any scientific method conducted for considering best treatment.

Answer: Thank you for the comment. The explanation for selecting these treatments was added to the text, as follows:

Line 507-515: “Based on the results of the previous analyses, the following was taken into account for selection of the best treatments for digestion assay: 1) adding maltodextrin to papaya seeds extracts improved their physicochemical properties and chemical stability, but in-creasing carrier amount from 15% to 30% had little impact on these changes; 2) the drying temperature had little effect on most evaluated parameters, so a lower temperature is more interesting for reducing costs and energy consumption; 3) bioactive release from particles containing unripe and ripe papaya seeds extracts has not been compared before and the release profile may be distinct due to their potentially different phenolic composition.”

Reviewer 3 Report

Some physicochemical properties of encapsulated papaya seed extract particles were evaluated. Study highlights the antioxidant potential of the metabolites present in the papaya seed, and the advantages of encapsulation to improve organoleptic properties and improve the stability of the metabolites.

Title: OK

Abstract: The objective described in the abstract does not match the title of the manuscript.

Introduction: OK

Materials and Methods: 1) The materials section is very concise, all the materials, equipment and reagents used in the investigation must be described in detail. Brand, origin, purity... 2) it is understood that the extracts obtained were solid, however line 81 mentions "extraction of oils".

Results: 1) Line 262. It is not possible to talk about solubility of extracts, the term solubility refers to the concentration of saturation of a certain substance... in this case there is no defined substance.

Conclusions: OK 

Author Response

Reviewer 3: Some physicochemical properties of encapsulated papaya seed extract particles were evaluated. Study highlights the antioxidant potential of the metabolites present in the papaya seed, and the advantages of encapsulation to improve organoleptic properties and improve the stability of the metabolites.

Title: OK

Abstract: The objective described in the abstract does not match the title of the manuscript.

Answer: The objective was modified as follows:

Line 13-15: “This study aimed to encapsulate papaya seeds extracts at different maturation levels, and to characterize the obtained microparticles for their physicochemical properties, chemical stability and release of bioactives.”

Introduction: OK

Materials and Methods: 1) The materials section is very concise, all the materials, equipment and reagents used in the investigation must be described in detail. Brand, origin, purity... 2) it is understood that the extracts obtained were solid, however line 81 mentions "extraction of oils".

Answer: 1) Detailed information about all reagents and equipment were included in the Materials and Methods section; 2) The mention to “extraction of papaya seed oil” in the cited line refers to the removal of oil from the seeds (considered an impurity in this case) to obtain defatted seeds, which were then used for production of the phenolic-rich extract.

Results: 1) Line 262. It is not possible to talk about solubility of extracts, the term solubility refers to the concentration of saturation of a certain substance... in this case there is no defined substance.

Answer: Thank you for the comment. The word “soluble” in the cited line was replaced by “dispersible”.      

Conclusions: OK

Round 2

Reviewer 1 Report

The authors improved the manuscript and it can be accepted in the present form

Reviewer 2 Report

The authors responded to all queries and modified the manuscript accordingly. 

Reviewer 3 Report

The suggestions and questions mentioned in report 1 were clarified and introduced correctly.